# Duplication, Divergence and Cardiac Expression of Tropoelastin in Jawed Fishes, Including Tetraploid Rainbow Trout (*Oncorhynchus mykiss*)

**DOI:** 10.3390/genes16121492

**Published:** 2025-12-13

**Authors:** Øivind Andersen, Tone-Kari Knutsdatter Østbye

**Affiliations:** 1Norwegian Institute of Food, Fisheries and Aquaculture Research (Nofima), 1433 Ås, Norway; tone-kari.ostbye@nofima.no; 2Department of Animal and Aquacultural Sciences, Norwegian University of Life Sciences (NMBU), 1433 Ås, Norway

**Keywords:** tropoelastin, extracellular matrix protein, hydrophobicity, subfunctionalization, bulbus arteriosus, Windkessel effect

## Abstract

Background/objectives: Tropoelastin is a highly hydrophobic extracellular matrix protein responsible for the extensibility and elastic recoil of various organs. The Windkessel effect in blood vessels dampens pressure variations during the cardiac cycle to provide continuous perfusion of tissues, such as the fragile gill capillaries in fish. The teleost-specific whole-genome duplication was followed by structural and functional divergence of the duplicated tropoelastins, of which ElnB confers the uniquely low stiffness of the bulbus arteriosus. Methods: We have examined the diversity of tropoelastins in all major fish clades by searching for *tropoelastin* (*eln*) genes in the sequenced genomes. Duplication of *eln* genes in tetraploid salmonids and cyprinids was examined by maximum likelihood phylogenetic analysis, and cardiac *eln* expression in rainbow trout was quantified by qPCR. Results: The tetraploid salmonid genomes harbor two *elna* genes but a single *elnb*, except for the tandem duplicated *elnb* genes in sockeye salmon and lake whitefish, while the tetraploid common carp possesses four *elna* and *elnb* genes on separate chromosomes. Rainbow trout showed strong elastin staining in the larval bulbus and ventral aorta, and the bulbar expression of *elnb* was 15 times higher than the ventricular levels in juvenile fish. The expression of *elna1* and *elna2* was also significantly higher in the bulbus, and together their transcript levels were almost similar as the *elnb* levels. The overall hydrophobicity of the fish tropoelastins differed considerably among the species ranging from 28.6% in Emerald rockcod ElnB to 56.3% in lesser devil ray Eln, but showed no significant difference with the tetrapods examined, except for the lower hydrophobicity of teleost ElnB. Conclusions: The inclusion of tetrapods in the analysis revealed a positive relationship between ventral aortic blood pressure and tropoelastin hydrophobicity.

## 1. Introduction

Elastin is an essential extracellular matrix protein in all jawed vertebrates by providing extensibility and elastic recoil in various organs like blood vessels, skin, lungs and ligaments. The elastin fibers consist of a microfibrillar mantle surrounding the insoluble elastin core, which is formed by self-assembly of the soluble tropoelastin monomers followed by covalent crosslinking of the elastin aggregates [1,2]. The extreme hydrophobicity, together with the multiple crosslinks, is pivotal for the insolubility, proteolytic resistance and exceptional stability and longevity of elastin with a half-time of about 70 years in man [2,3]. The tropoelastin protein consists of alternating hydrophobic and hydrophilic domains, which usually correspond to the individual exons in the genetic sequence [4]. The hydrophobic domains interact in the self-assembly, or coacervation, process and are particularly rich in non-polar amino acids typically occurring in repeated motifs [5,6,7,8]. The hydrophilic crosslinking domains are subdivided into KA and KP types, which consist of alanine and proline residues, respectively, between pairs of lysine and form highly stable crosslinks by the action of lysyl oxidases [4,9,10]. The KA and KP types can be covalently crosslinked with each other, and substitution of KP for KA crosslinks had little effect on the tensile mechanical properties of elastin-like proteins [10,11]. The conserved C-terminal end of tropoelastin contains a tetrabasic motif and a cysteine pair of importance for the interactions with other matrix components [12,13,14].

The elasticity of blood vessels is crucial for smoothing the pulsatile blood flow generated by the heart to provide continuous perfusion of peripheral tissues [15]. The large amounts of elastin fibers in the aortic wall create the arterial elasticity below and at physiological pressure, while the collagen fibers provide arterial stiffness to withstand the force of high blood flow [16]. The expanding energy from the blood pressure is temporarily stored in the elastic arteries when the walls are stretched during the systolic phase and is returned to the blood flow when the elastin recoils during diastole [17]. This “Windkessel” effect also occurs in the elastic bulbus arteriosus of the teleost heart, acting as a damping chamber to provide a more even flow through the gill lamellae for efficient gas exchange and for protecting the delicate gill capillaries [18,19,20,21,22]. The teleost-specific whole-genome duplication (WGD) event about 300 million years ago generated the duplicated *elna* and *elnb* genes, which exhibit divergent expression patterns and functions in zebrafish [23,24]. The ElnB paralog plays a key role in the development of the bulbus by promoting the differentiation of cardiac precursor cells into smooth muscle [23,24]. Zebrafish ElnB was recently suggested to confer the considerably lower stiffness of the bulbus than the ventricle, in contrast to the high stiffness in both the muscular conus arteriosus and ventricle in the non-teleost ray-finned gray bichir (*Polypterus senegalus*) [25]. Consistently, the extended hydrophobic domains in zebrafish ElnB were predicted to have implications for the decreased elastic modulus, assuming a rubber model for the elasticity of elastin [4,23]. The divergence and subfunctionalization of the duplicated tropoelastins should be further examined in non-model fish species, including tetraploid salmonids and cyprinids. Multiple gene paralogs retained from the salmonid WGD were identified in the sequenced genomes from various subfamilies [26]. The few molecular studies of salmonid tropoelastin include the extraction of a single *eln* mRNA from the bulbus in coho salmon (*Oncorhynchus kisutch*), while the immunoreactive protein of 43 kD was considerably smaller than the 70 kD tropoelastin in mammals [27].

The mechanism underlying the reversible elasticity of elastin has been controversial for decades and was recently concluded to be primarily driven by the hydrophobic effect that was suggested to account for elastin’s low stiffness and high resilience [28]. An evolutionary trend towards increased overall hydrophobicity in vertebrate tropoelastin was proposed to be an adaptive advantage in homeothermic vertebrates related to the higher blood pressure in the more advanced circulatory systems [29,30,31], or by lowering the coacervation temperature to facilitate elastin formation in mammals, birds and alligators [32]. The structural and functional properties of elastin should be investigated in a broad range of fish species to better understand the evolution of this unique protein and the relationship with different physiological adaptations and functional requirements among species. Many fish species live in extreme environments, such as Antarctic icefish permanently inhabit ice-cold water and are apparently lacking bulbar elastin [21]. Deep-sea snailfish (*Pseudoliparis swirei*) has developed unique adaptations to withstand the extreme hydrostatic pressure at depths below 8000 m in the Mariana trench [33,34]. The Greenland shark (*Somniosus microcephalus*) has the longest lifespan in any known vertebrate, of several hundred years, which may be reflected in elastin stability and longevity. Here, we examine the diversity of fish elastin in a variety of species spanning about 450 million years of evolution. Elastin gene duplications of the paralogs were investigated in tetraploid cyprinids and salmonids, including rainbow trout (*Oncorhynchus mykiss*) possessing three *eln* genes, which were differentially expressed in bulbus and ventricle.

## 2. Materials and Methods

### 2.1. Identification and Characterization of Fish Elastins

To examine the diversity of elastin in jawed fish, we selected 51 species differing in living habitat, swimming activity and cardio-vascular morphology. We searched for *eln* genes in the sequenced fish genomes available at NCBI’s Genome Resources (https://www.ncbi.nlm.nih.gov/home/genomes (accessed on 1 November 2025)) by comparing nucleotide (BLASTN) and protein sequences (BLASTP) (https://blast.ncbi.nlm.nih.gov/Blast.cgi (accessed on 1 November 2025)). Uncharacterized or misannotated or uncharacterized *eln* genes were identified by searching for the syntenic genes *limk1* and *septin4*/*5* [35], which were conserved in all species examined, except for the missing *elnb* in the tarpon genome (Appendix A). All tropoelastins were examined for full-length size by inspecting the N-terminal signal peptide (https://services.healthtech.dtu.dk/services/SignalP-6.0/ (accessed on 1 November 2025)) and the basic Cys-containing C-terminus. Partial or missing *eln* genes were verified by screening the genome using TBLASTN at NCBI, while short *eln-like* sequences identified in the incomplete cyprinid genomes were excluded. The teleost ElnA and ElnB paralogs were distinguished by the extended exons and lower hydrophobicity of ElnB than ElnA. In species where only a single paralog was identified, the identity was verified by blasting the protein sequence against the ElnA and ElnB paralogs from related species.

A phylogenetic tree of the tropoelastins in tetraploid salmonid and cyprinid species was made by alignment of the amino acid sequence data using the ClustalW algorithm implemented in the software package MEGA (version 12) [33]. MEGA was also used to perform a maximum likelihood phylogenetic analysis with a Jones–Taylor–Thornton (JTT) substitution model and uniform rates among sites. Heuristic tree searching was performed using the Nearest-Neighbor-Interchange (NNI) method, and bootstrapping with 100 pseudoreplicates was used to assess tree robustness.

The overall hydrophobicity of full-length tropoelastin proteins was calculated using the Kyte-Doolittle hydrophobicity scale [36] (https://www.peptide2.com/N_peptide_hydrophobicity_hydrophilicity.php (accessed on 1 November 2025)). The Kyte–Doolittle hydropathy analysis of rainbow trout ElnB was performed by ProtScale (https://web.expasy.org/protscale/ (accessed on 1 November 2025)) with a sliding window size of 9 amino acids. Repeated hydrophobic motives and crosslinks were identified by the ScanProsite tool (https://prosite.expasy.org/scanprosite/ (accessed on 1 November 2025)). Graphpad Prism 10.4.1 was used to visualize the overall hydrophobicity (%) of the tropoelastins and to compare tropoelastin hydrophobicity and mean ventral aortic blood pressure. Due to the small sample size, the hydrophobicity data did not meet the assumption for normality using the D’Agostino and Pearson test, and one-way ANOVA followed by Kruskal–Wallis test was therefore applied.

### 2.2. Fish Hold and Heart Sampling

Rainbow trout eggs were fertilized with milt of sex-reversed females supplied by AquaGen breeding company and incubated at 10 °C at the Aquaculture Research Station at Sunndalsøra, Norway, from 22 March 2023. The hatched larvae were transported before starting to feed to Svanøy Havbruk farming rainbow trout at Svanøy island on the west coast of Norway. The all-female fish were raised at 10 °C under 24 h light cycle that was changed to 12 h:12 h light-dark cycle at 100 g to adapt the fish to natural photoperiod before seawater transfer. The fish were kept in fiberglass tanks from 16 to 280 m^3^ at densities varying with the fish size and were supplied with water from the local lake, Sagevatn, at water flow and stock densities varying with fish age. The oxygen levels were monitored continuously, and the fish were fed daily with commercial pellets adjusted to the appetite using automated feeders. Pit-tagged fish were transferred to seawater on 15 December 2023 and were kept in net pens until slaughtering at body weight of about 3500 g on 12 December 2024.

Twenty fish were sampled for qPCR analysis of the cardiac expression of the three *eln* genes during the freshwater phase at body weight of about 1 g (22 May 2023), 10 g (18 August 2023), and 100 g (1 December 2023), and finally at about 700 g in seawater (13 May 2024). The four time points for tissue sampling throughout one year were chosen for proper removal of the heart from 1 g fry and for dissection of the bulbus and ventricle from 10 g juveniles and later from 100 and 700 g fish before and after seawater transfer. The fish were anesthetized with MS-222 (tricaine methanesulfonate, 150 mg·L^−1^) and killed with a blow to the head at Svanøy Havbruk. The heart was removed and was stored in RNA-later at −80 °C for four months until extraction of RNA from bulbus and ventricle dissected under microscope. Ten fry of 1 g were fixed in 10% PFA and kept at room temperature before histology and staining.

The study was conducted in accordance with the European Union Directive 2010/63/EU and the National Guidelines for Animal Care and Welfare established by the Norwegian Ministry of Education and Research. The Norwegian Food Safety Authority approved the experiment (FOTS ID 29083). Key personnel involved in the fish trial held FELASA C certification.

### 2.3. Histology

Formalin-fixed fry (*n* = 10) were processed overnight in Tissue Processor (Logos EVO, Milestone, Italy). Paraffin-embedded whole fry were sectioned (2 μm) in sagittal plan using a rotary microtome (Leica RM2255, Biosystems, Muttenz, Switzerland) and stained with Elastin Van Gieson (EVG) staining kit (Atom Scientific Ltd., Hyde, UK) according to manufacturer’s instructions. The slides were then analyzed using light microscopy and the QuPath0.3.2, (Quantitative Pathology & Bioimage Analysis) software. Images were scanned with Leica AperioCS2 slide scanner (Leica Biosystems, Deer Park, IL, USA) at 40× magnification.

### 2.4. Gene Expression Analysis

Total RNA was extracted from heart tissue using Proteinase K digestion followed by Agencourt RNAadvanced Tissue Kit^®^ (Beckman Coulter Inc., Brea, CA, USA) using the Biomek 4000^®^ robotic workstation according to the manufacturer’s protocol. RNA concentration and purity were assessed using a Nanodrop 8000 Spectrophotometer (Thermo Fisher Scientific, Waltham, MA, USA). Complementary DNA (cDNA) was synthesized from mRNA using the Qiagen QuantiTect Reverse Transcription Kit^®^ that includes genomic DNA removal (Qiagen, Valencia, CA, USA) and then diluted 1:10 for qPCR analysis. No enzyme control and no template control were included as negative controls. The PCR mix consisted of 0.5 μL forward primer (10 μM), 0.5 μL reverse primer (10 μM), 5 μL PowerUp SYBR green (Thermo Fisher Scientific, USA) and 4 μL diluted cDNA. The qPCR conditions were: 50 °C/2 min, 95 °C/20 s; 40 cycles of 95 °C/1 s, 60 °C/20 s. Duplicate technical replicates were included in the qPCR analysis. Biological replicates consisted of 8 samples for bulbus at 10 g, 9 samples for ventricle at 100 g, 10 samples for ventricle at 10 g and 700 g, and 10 samples for bulbus at 100 g and 700 g. The melting curve conditions were 95 °C/1 s, 60 °C/20 s, 95 °C/1 s. Specificity of all primers (Thermo Fisher Scientific, MA, USA) was confirmed by Sanger Sequencing (Eurofins Genomics; Ebersberg, Germany) of amplicons (Table 1). *ef1a* and *b-act* were evaluated as reference genes using RefFinder [37], and the relative gene expression level was calculated according to the ΔΔCt method [38] using *ef1a* as reference gene. Normality was assessed using the Shapiro–Wilk test, which indicated that the data for *elna2* and *elnb* were normally distributed (*p* > 0.05), but not the data for *elna1*. Examination of the homoscedasticity plot showed that *elna1* and *elna2* gene expression data met the assumption of homoscedasticity, whereas *elnb* did not. Since both normality of the data and homogeneity of the variance were satisfied for *elna2*, statistical significance was assessed by using one-way ANOVA followed by Tukey’s post hoc test (*p* < 0.05). For *elnb* and *elna1* gene expression, one-way ANOVA with Kruskal–Wallis test was applied.

## 3. Results

We identified up to four different *eln* genes in the sequenced genomes from 51 jawed fish species, including cartilaginous fishes, basal ray-finned fishes, teleost fishes, and lobe-finned fish (Appendix A). All genomes examined were shown to contain a syntenic block comprising *eln*, *septin4*/*septin5* and *limk1*, except for the missing tarpon (*Megalops cyprinoides*) *elnb* linked to *septin5* and *limk1* on chromosome 3. Examination of the jawless fish genomes revealed no *eln*-like gene close to the hagfish *septin5* (LOC137430108), lamprey *septin4* (LOC116957768) and lamprey *limk1* (LOC116958416).

All non-teleost genomes contain a single *eln* gene, whereas duplicated *elna* and *elnb* genes were found in teleosts, including Northern pike (*Esox lupius*) and grass carp (*Ctenopharyngodon idella*), which are diploid relatives to tetraploid salmonids and cyprinids, respectively. The salmonids examined have duplicated *elna1* and *elna2* paralogs, but only a single *elnb* gene, except for the tandem duplicated *elnb1* and *elnb2* genes in sockeye salmon (*Oncorhynchus nerka*) and lake whitefish (*Coregonus clupeaformis*) (Figure 1). Common carp has four *eln* genes named *elna1*, *elna2*, *elnb1* and *elnb2* located on separate chromosomes, while only *elna1*, *elna2* and *elnb* were identified in goldfish (*Carassius auratus*).

All fish *eln* genes were found to contain multiple hydrophilic and hydrophobic exons, but the exon length varies considerably among the species and between the paralogs. Probably the most remarkable genetic structure is found in the lesser devil ray (*Mobula hypostoma*) *eln* gene, which consists of 70 exons containing multiple copies of 36-nt and 48-nt long exons coding for alternating hydrophilic and hydrophobic domains, respectively (Figure 2 and Appendix A). The lesser devil ray tropoelastin has 24 KAAK crosslinks, and the KA type also dominated in teleost ElnA, including rainbow trout ElnA1 and ElnA2 (Figure 3A,B). Most ElnB paralogs contained both KA and KP types, such as rainbow trout ElnB (Figure 3C), while the ray-finned species spotted gar, bowfin and gray bichir exhibited mainly KP domains (Appendix A). Similarly to the single *eln* gene in non-teleost fish, the teleost *elna* gene comprises multiple short exons, whereas the *elnb* paralog consists of considerably larger hydrophobic and hydrophilic exons. For example, yellowfin tuna (*Thunnus albacares*) ElnA of 1096 aa has 52 short exons, except for the 180-nt long exon 20, while the partial ElnB of 1433 aa is coded by 31 extended exons, such as 441-nt long exon 28. The genetic structure also differs between the *elna* and *elnb* genes in the tetraploid salmonids and cyprinids. The three rainbow trout *eln* genes have about the same number of exons, but the 2441-aalong ElnB consists of much longer hydrophobic and hydrophilic domains compared to ElnA1 and ElnA2 of 1357 and 1403 aa, respectively (Figure 3). KA and KP crosslinks are found in both short and long domains of ElnB, making the distinction between hydrophobic and crosslinking regions less clear. The hydrophobicity plot of rainbow trout ElnB shows that the extended domains comprise both hydrophobic and hydrophilic sequences, consistent with the low overall hydrophobicity of ElnB (Appendix A).

Histochemical analysis of 2-month-old rainbow trout fry revealed strong elastin staining in the bulbus and ventral aorta (Figure 4). Elastin was weakly stained in the bulboventricular valve and pericardium, but not in the ventricle. qPCR quantification of the three *eln* gene expression levels showed that *elnb* dominated in the fry heart, and the *elnb* expression was 15 times higher in the bulbus than in the ventricle of the juvenile fish (Figure 5). The bulbar expression of *elna1* and *elna2* genes was significantly lower than *elnb*, but the total amounts of *elna1* and *elna2* transcripts were almost similar to the *elnb* transcripts in the bulbus.

The overall hydrophobicity of fish tropoelastin varies largely among species, ranging from 28.6% in Emerald rockcod ElnB to 56.3% in the lesser devil ray Eln. The hydrophobicity of the teleost ElnB paralogs (33.4 ± 3.6%) was significantly lower than the levels in cartilaginous fish (48.7 ± 3.8%) and teleost ElnA (42.2 ± 3.9%), which were not significantly different from the hydrophobicity of the reptiles, birds and mammals examined (Figure 6).

A relation between the mean ventral aortic blood pressure and the overall tropoelastin hydrophobicity was examined in various fish species. No correlation was found between the blood pressure levels and the hydrophobicity levels of non-teleost Eln and teleost ElnA (Figure 7). Moreover, the overall hydrophobicity of the teleost ElnB paralog showed no relationship with the blood pressure in the bulbus, where pressure levels were derived from the strong correlation between bulbar plateau pressure and ventral aortic pressure [39]. By broadening the analysis to include tetrapods, we identified a positive relationship between ventral aortic pressure and tropoelastin hydrophobicity (Figure 7).

## 4. Discussion

The systematic survey of *eln* genes in the sequenced genomes from diverse fish species identified up to four different tropoelastins in jawed species sharing the characteristic properties of multiple alternating hydrophobic and hydrophilic domains with KA and KP crosslinks ending with a Cys pair in the negatively charged C-terminal end. Whereas the linear sequence of the single tropoelastin in sharks and rays seems to be well conserved, teleost ElnA and ElnB have diverged considerably in accordance with the subfunctionalization of the paralogs in zebrafish [23,24,25]. Consistent with the differential expression of *elna* and *elnb* in the developing zebrafish heart, the expression of rainbow trout *elnb* dominated in the fry heart and in the bulbus at later stages. The spatial and temporal expression of *elnb* should be further compared with *elna1* and *elna2* in the developing heart during embryogenesis using in situ hybridization and specific antibodies.

The low stiffness imparted by zebrafish ElnB for proper bulbus cell fate and function was predicted to be the result of the long hydrophobic domains in ElnB [4,23,25]. The high distensibility and resilience of the bulbus have consistently been reported in tuna fish, carp and rainbow trout [4,23,25] and were calculated to account for 25% of the blood flow in rainbow trout at rest [40]. Thermal remodeling of the rainbow trout heart during cold acclimation resulted in decreased elastin-to-collagen ratio and increased stiffness in the bulbus [41]. Intriguingly, the *elnb* gene is apparently missing in the genome of the tarpon, which branched off very early in teleost evolution. This athletic air-breathing fish has a unique heart with a prominent bulbus arteriosus in addition to a distinct, vestigial conus arteriosus [42,43]. The highly hydrophobic tarpon ElnA consists of only short hydrophobic and hydrophilic domains and probably does not function like an ElnB paralog in this species.

The Antarctic icefish ElnB paralog has very low overall hydrophobicity, such as the red-blooded Emerald rockcod (*Trematomus bernacchii*), which, together with the white-blooded crocodile icefish (*Chionodraco hamatus*), lacks elastin in the bulbus [44,45]. The low hydrophobicity and few crosslinks may explain the possible inability of the bulbar elastin in Antarctic icefish to aggregate into larger units at freezing temperatures [44]. In human elastin, disruptions of the hydrophobic domains were shown to be detrimental to the self-assembly process, and mutated human elastin required much higher temperatures than the normal physiological temperature to achieve full coacervation [46]. On the other hand, the absence of elastin fibers in Antarctic fish could be simply the result of extreme morpho-functional adaptation to constant sub-zero temperatures [46]. The Antarctic icefishes have very large hearts, high blood volumes and low blood pressure, and smooth muscle cells may contribute to the elastic properties of the bulbus, maintaining constant aortic flow in the large branchial vessels [45,47,48,49].

Except for Antarctic icefishes, the various fish species examined revealed no relationship between tropoelastin hydrophobicity and blood pressure in ventral aorta. Cartilaginous fish exhibit highly hydrophobic tropoelastin, but slow-mowing sharks have low blood pressure below 4 kPa, while blood pressure around 7 kPa has been recorded in fast-mowing sharks, which is probably similar in rays [50,51,52,53]. In comparison, lobe-finned fish exhibited relative high tropoelastin hydrophobicity, while low blood pressure of 3.1 kPa was measured in West-African lungfish (*P. annectens*) [54]. Similarly, the sterlet sturgeon (*Acipenser ruthenus*) showed the highest hydrophobicity of 44.8% among non-teleostean ray-finned fish, but blood pressure less than 3 kPa was reported in white sturgeon [55]. Except for tarpon and baby whalefish (*Brienomyrus brachyistius*), all teleost tropoelastins examined have hydrophobicity below 50%, but show large differences in ventral aortic blood pressure varying from 0.3 to 13 kPa in zebrafish and tuna fish, respectively [56,57]. While the overall hydrophobicity of ElnA is higher than ElnB in the two species, the importance of the long hydrophobic domains for the elasticity in zebrafish ElnB [4,23] should be further investigated in other teleost species. On the other hand, the fast-swimming tuna fish has thick aortic elastin lamella and bulbar elastin fibers, in contrast to the thinner, less developed elastin fibers in zebrafish [22,23,57]. Consistently, the higher elastin content in some species than in others probably gives higher tissue compliance and elasticity [19,21,40,58,59].

## 5. Conclusions

The single tropoelastin in non-teleosts has been well conserved during 450 million years of evolution, but structural differences among species may be related to dissimilar adaptations and functional requirements. The structural and functional divergence of the duplicated ElnA and ElnB seems to be similar in zebrafish and tetraploid rainbow trout, but the cardiac expression of the three *eln* genes in rainbow trout should be further examined during embryogenesis. The crucial role played by ElnB in the development of bulbus seems to be absent in Antarctic icefish, lacking bulbar elastin, and in the early teleostean tarpon, apparently missing the *elnb* gene. This study showed no evolutionary tendency towards increased hydrophobicity of vertebrate tropoelastins. When tetrapods were incorporated into the analysis, a positive association emerged between ventral aortic pressure and tropoelastin hydrophobicity.

## Figures and Tables

**Figure 1 genes-16-01492-f001:**
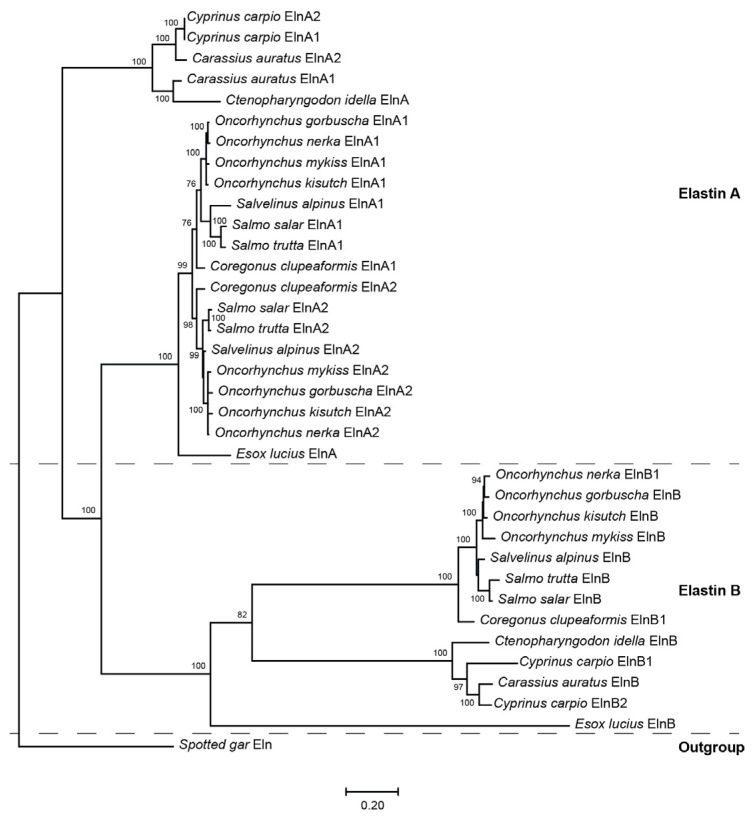
Maximum likelihood phylogenetic analysis of the tropoelastins in tetraploid salmonids and cyprinids based on amino acid sequences. The tandem duplicated ElnB2 in sockeye salmon and lake whitefish was excluded due to low sequence similarities. Northern pike and grass carp ElnA and ElnB were included for comparison, and spotted gar Eln was used as outgroup. Bootstrap values > 75% have been indicated. NCBI accession numbers are given in Appendix A.

**Figure 2 genes-16-01492-f002:**
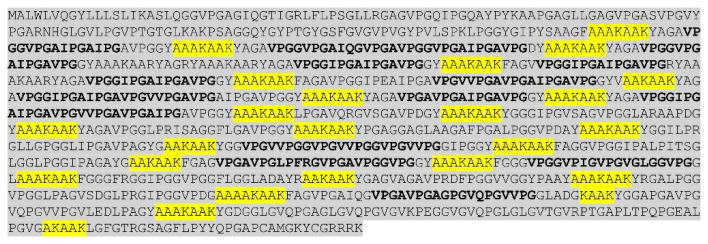
Elastin of the lesser devil ray with alternating hydrophilic and hydrophobic domains. Crosslinks and preceding Ala residues are in yellow color, and imperfect repeats of hydrophobic motives shown in bold. The multiple copies of the corresponding hydrophilic and hydrophobic exons are given in Appendix A.

**Figure 3 genes-16-01492-f003:**
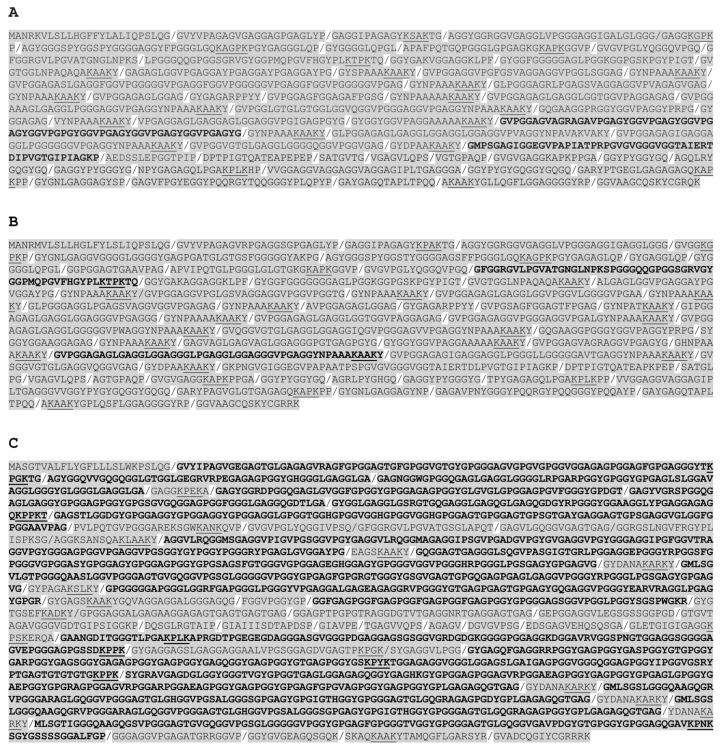
Amino acid (aa) sequences with exon borders (/) of rainbow trout ElnA1 (**A**), ElnA2 (**B**) and ElnB (**C**). Exons coding for more than 50 aa are shown in bold. The KA and KP types of crosslinks are underlined.

**Figure 4 genes-16-01492-f004:**
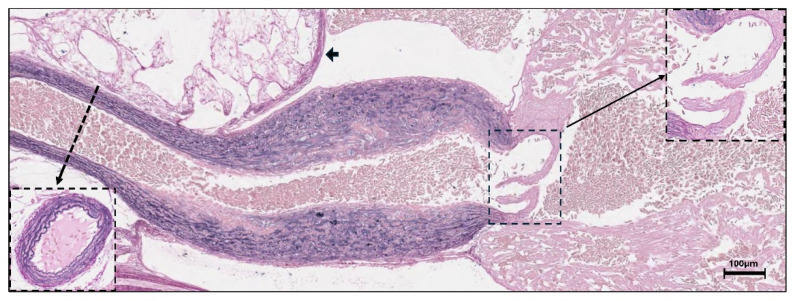
Histochemical EVG staining of elastin in the heart and ventral aorta of rainbow trout fry at two months. Cross-section of ventral aorta and enlarged image of bulboventricular valve are included. Arrowhead shows elastin in pericardium.

**Figure 5 genes-16-01492-f005:**
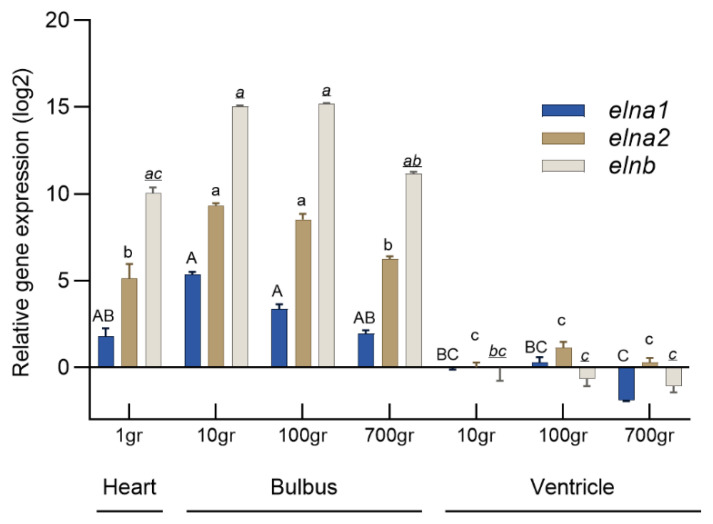
qPCR quantification of relative expression levels (mean ± SE) of rainbow trout *elna1*, *elna2* and *elnb* in bulbus and ventricle at 1, 10, 100 and 700 g body weight (*n* = 8–10). The relative expression levels were normalized using *ef1* as reference gene. Different letters indicate significant differences *elna1* (uppercase letters), *elna2* (lower case) and *elnb* (lower case and underlined).

**Figure 6 genes-16-01492-f006:**
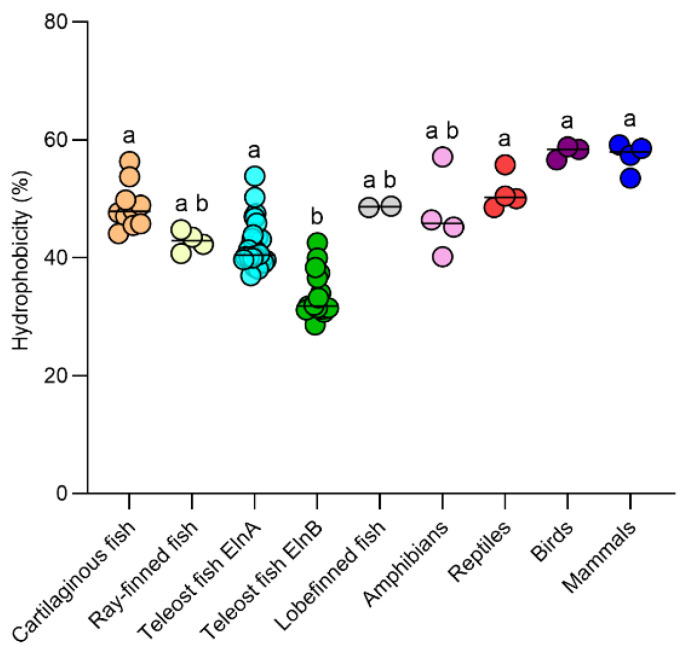
The overall hydrophobicity (Kyte–Doolittle) of full-length tropoelastins in the major fish clades and selected tetrapod species shown in different colors and with the median indicated. Different letters indicate significant differences.

**Figure 7 genes-16-01492-f007:**
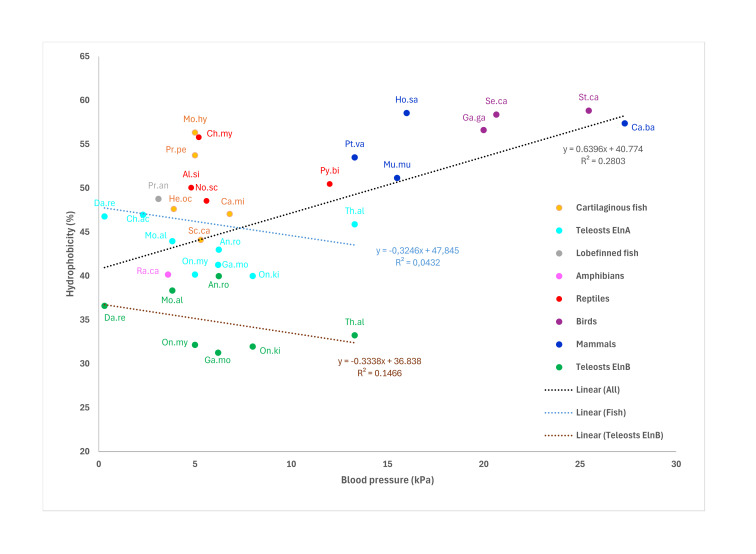
Tropoelastin hydrophobicity plotted against mean ventral aortic blood pressure in various fish and tetrapod species. Trend lines including equation and R^2^ are given for all species (black), for all fish species (blue), and for teleosts ElnB (brown). References to the blood pressure measurements are given in Appendix A. The codes refer to the scientific names. He.oc: Epaulette shark (*Hemiscyllium ocellatum*), Ca.mi: Elephant shark (*Callorhinchus milii*), Sc.ca: Spotted catshark (*Scyliorhinus canicula*), Mo.hy: Lesser Devil ray (*Mobula hypostoma*), Pr.pe: Smalltooth sawfish (*Pristis pectinata*), Mo.al: Swamp eel (*Monopterus albus*), Da.re: Zebrafish (*Danio rerio*), Ch.ac: Blackfin icefish (*Chaenocephalus aceratus*), Th.al: Yellow-fin thuna (*Thunnus albacares*), Ga.mo: Atlantic cod (*Gadus morhua*), An.ro: American eel (*Anguilla rostrata*), On.my: Rainbow trout (*Oncorhynchus mykiss*), On.ki: Coho salmon (*Oncorhynchus kisutch*), Pr.an: West-African lungfish (*Protopterus annectens*), Ra.ca: American bullfrog (*Rana catespiana*), Ch.my: Green sea turtle (*Chelonia mydas*), No.sc: Tiger snake (*Notechis scutatus*), Al.si: Chinese alligator (*Alligator sinensis*), Py.bi: African python (*Python bivittatus*), Ga.ga: Chicken (*Gallus gallus*), Se.ca: Common canary (*Serinus canaria*), St.ca: South African ostrich (*Struthio camelus*), Ho.sa: Human (*Homo sapiens*), Mu.mu: Mouse (*Mus musculus*), Pt.va: Large flying fox (*Pteropus vampyrus*), Ca.ba: Bactrian camel (*Camelus bactrianus*).

**Table 1 genes-16-01492-t001:** Primers used in the qPCR analysis of rainbow trout *eln* and reference genes.

Gene	Genbank Id	Primer Sequence (5′-3′)	Efficiency
*elna1*	XM_036965318.1	F: ttcgatactgctctggcatgt	1.96
		R: tggcccctaatctagcacac	
*elna2*	XM_021582450.2	F: tgtagcctactccgtgatggt	1.99
		R: cggtattgctgggcacaagt	
*elnb*	XM_036944404.1	F: caaatcaggttatggctcctcct	2.00
		R: tgcatggctgtgtatttggct	
*ef1a*	AF498320.1	F: attaacattgtggtcattggccatgtc	2.03
		R: atctcagctgcttccttctcgaactttt	
*b-act*	XM_036973727.1	F: ggaggctccatcttggcttc	2.00
		R: gaagtggtagtcgggtgtgg	

## Data Availability

The original contributions presented in this study are included in the article and in Appendix A. Further inquiries can be directed to the corresponding author. Appendix A gives the accession numbers of the fish elastin examined. Appendix A gives the alignment of the multiple copies encoding alternating hydrophilic and hydrophobic domains in the lesser devil ray elastin. Appendix A shows ray-finned and lobe-finned tropoelastin protein with crosslinks. Appendix A gives the Kyte-Doolittle hydropathy plot of rainbow trout ElnB.

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
