# Peer review of "Duplication, Divergence and Cardiac Expression of Tropoelastin in Jawed Fishes, Including Tetraploid Rainbow Trout (Oncorhynchus mykiss)"

_genes, 2025, doi:10.3390/genes16121492_

Round 1

Reviewer 1 Report

Comments and Suggestions for Authors

Referee Report on the Manuscript: “Duplication and Divergence of Elastin in Jawed Fishes, Including Tetraploid Rainbow Trout (Oncorhynchus mykiss)” submitted to the “Genes” periodical.

n the reviewed study, the authors performed a comparative genomic, structural, phylogenetic, and expression analysis of elastin (eln) genes across 51 jawed fish species, including cartilaginous fishes, basal ray-finned fishes, teleosts, and lobe-finned fishes. The study offers several original contributions, some of which have not been previously reported, including:

  • The most comprehensive survey to date of elastin gene duplication and structure across jawed fishes
  • A new evolutionary interpretation of elna / elnb division in teleosts
  • The discovery of extreme exon amplification in the lesser devil ray elastin gene
  • Clarification of paralog retention after lineage-specific tetraploidization
  • Integration of structural protein hydrophobicity with physiological data like blood pressure
  • The first expression comparison of all three eln genes (elna1, elna2, elnb) in rainbow trout heart

Therefore, the presented findings substantially advance our understanding of elastin evolution, particularly in teleosts and tetraploid salmonids, and uncover unexpected structural diversity that may stimulate new biomechanical or developmental research.

The introduction is largely accurate in its description of elastin structure, assembly, biochemical properties, and cardiovascular function across vertebrates. Foundational statements about elastin domains, crosslinking, hydrophobicity and self-assembly are consistent with classic and modern literature. However, important evolutionary statements are oversimplified or insufficiently referenced, some claims are speculative, some terminology is misleading and the narrative is not enough tightly connected to the study’s aims:

  1. the phrase “tropoelastin monomers (onwards named elastin)” is misleading, because tropoelastin and elastin are distinct: tropoelastin is the soluble precursor, and elastin is the insoluble, crosslinked polymer;
  2. The statement that elastin is present “in all jawed vertebrates” needs nuance and references. It is true for most gnathostomes, but absence of bulbar elastin in Antarctic icefish (cited later) contradicts this generalization. Authors should to acknowledge known exceptions immediately, not many lines later;
  3. The claim that the bulbus arteriosus emerged after WGD ~300 Mya is in my opinion oversimplified as current comparative anatomical literature suggests that a bulbus arteriosus-like structure exists in basal actinopterygians (gars, bowfin), not only in teleost and its evolution is not clearly tied to the teleost WGD event. This statement requires updated references or revision;
  4. At present, the claim that duplicated eln genes “resulted in neofunctionalization of ElnB” is rather speculative as only a few studies (largely zebrafish-focused) discuss functional divergence. “Neofunctionalization” is a strong claim and should be toned down without functional studies across taxa;
  5. Evolutionary claims about hydrophobicity and endothermy require stronger referencing. Also, statements linking hydrophobicity to high blood pressure or coacervation temperature depend heavily on a few speculative papers. I suggest to expanded referencing and keep more cautious wording (e.g., “proposed” instead of “suggested to be an adaptive advantage”);
  6. The introduction does not clearly justify why studying elastin diversity in 51 fish is important beyond broad evolutionary interest. Specifically, missing elements include: (1) explanation of knowledge gaps (e.g., limited comparative studies of eln in non-model fishes), (2) how exon-level diversity may influence biomechanics and (3) why tetraploid species are especially informative;
  7. There is lack of any mention of functional studies on salmonid elastins (if available) and Clear integration of whole genome duplication theory relevant to salmonids and cyprinid;
  8. Overall, English is understandable and only requires minor editing.

The material and methods chapter is generally logical (comparative genomics + phylogenetics + histology + qPCR) and appropriate for the study’s aims. However, the description of the methods sometimes seems to be incomplete and vague:

  1. No description of inclusion/exclusion criteria for selecting genomes (51 species is mentioned in the Results but not in Methods);
  2. No explanation of genome assembly quality (many fish assemblies are fragmented or contain misannotations);
  3. No quality control of gene predictions (manual curation? blastx? reciprocal BLAST?);
  4. Hydrophobicity analysis lacks replicability (no sliding-window parameters, no smoothing, no amino acid scales besides Kyte–Doolittle);
  5. BLAST method not specified: BLASTP? BLASTN? TBLASTN?
  6. Search parameters missing (e-value threshold, identity %, coverage);
  7. No description of how partial or fragmented hits were handled;
  8. No pipeline for distinguishing elnA vs. elnB paralogs;
  9. Synteny analysis only states “adjacent limk1 and septin4” but gives no method for validating synteny (genome browser? NCBI? Ensembl?;
  10. Manual search for motifs” is vague and not acceptable as a reproducible bioinformatic procedure;
  11. Presently, it is considered that NJ method is outdated. Authors should justify its use or preferably use maximum likelihood (ML) in my opinion;
  12. For phylogenetic analyses there is no information about alignment trimming, gaps treatment, nucleotide substitution model selection and whether elnB2 sequences were excluded a priori;
  13. For hydrophobicity analysis the link to an online calculator is insufficient for reproducibility. Missing information about window size Model justification, whether the full-length proteins were aligned or compared positionally;
  14. Fish Holding and Sampling subchapter:
  • only 20 fish per timepoint are mentioned, but there is no replication structure (biological vs. technical replicates), sex information, or justification for sample size;
  • No explanation why specific timepoints were chosen in relation to elastin biology;
  • The research claims to investigate tetraploid salmonids and cyprinids, but only rainbow trout is sampled for physiology and expression;
  • temperature, oxygen, light regime too coarse; e.g., “constant light” is inappropriate for salmonids and should be justified;
  • Growth conditions (tank densities, feeding rate, water flow) are missing;
  • No description of heart dissection protocol, RNA-later incubation time, or tissue storage temperaturÄ™;
  • The larvae were “1 g” – rather “1 g fry”, not larvae;
  • Transport, stress handling, and acclimation periods before sampling not described.
  1. Histology subchapter missing details on:
  • No mention of the number of sections analyzed per fish, the anatomical orientation, or blinding of histological assessments
  • Fixation time in paraformaldehyde;
  • Orientation of embedding;
  • Sectioning plane;
  • Image acquisition settings (magnification, camera, scale bars);
  • Staining controls
  1. Molecular Analysis subchapter missing details on:
  • Subchapter title – better fits gene expression analysis or just “Real-time PCR analysis”;
  • RNA integrity not assessed;
  • No description of DNAse treatment;
  • SYBR Green qPCR conditions are oversimplified and lack of exact replicate numbers (technical & biological);
  • There is no explaination of statistic test selection and checking of data distribution and variance homogeneity.

The results appear generally clear, but some sections could benefit from more explicit highlighting of trends, clearer linkage between figures/tables and text, and more precise explanations of key comparisons:

  1. Paragraphs frequently mix results with interpretation;
  2. Missing clarity about assembly quality (which influences gene detection);
  3. Statements such as “considerably larger” are qualitative and actual exon lengths should be reported;
  4. No statistical values shown for gene expression analysis results (fold change, p values);
  5. The statement “The bulbus and ventricle were not dissected in the larvae” appears misplaced and breaks narrative flow;
  6. “up to 15 times higher” is vague—use exact fold-change per timepoint;
  7. Claims like “varies largely” or “highly hydrophobic elastin was consistently found” need statistical support;
  8. Hydrophobicity values should be given as mean ± SD;
  9. It is unclear whether analyses used full-length sequences, ortholog alignments, or paralog-specific datasets;
  10. Relationship with blood pressure – no numeric correlation coefficient (r, p-value);
  11. “In comparison, the tetrapod species examined possessed high elastin hydrophobicity and high blood pressure levels.” → observational but should be supported visually and numerically;
  12. No statistical test described for relationship with blood pressure (linear regression? Spearman?).

The Discussion is requires restructuring, clarification, and moderation of speculative claims:

  1. Statements like “high distensibility and resilience of the bulbus have consistently been reported in various teleost species” are broad. No specific numbers or citations for all species mentioned;
  2. Claims about snailfish Glu/Asp content affecting vessel stiffness is rather speculative as no functional or experimental data supports this;
  3. Linking hydrophobicity directly to blood pressure across species is overstated, given results showed no correlation;
  4. The functional significance of Antarctic icefish adaptations is partially conjectural without experimental validation;
  5. Paragraphs mix multiple topics (e.g., gene divergence, expression patterns, hydrophobicity, ecological adaptations) without smooth transition;
  6. The flow jumps between rainbow trout, Antarctic fish, teleosts, and mammals abruptly, making the argument hard to follow;
  7. Use of “athletic tuna fish has thick aortic elastin lamella and bulbar elastin fibres” lacks quantification.

The conclusions are partially supported by the results: they correctly note the conservation of non-teleost elastin and divergence of teleost ElnA/ElnB, but they omit key findings (e.g., hydrophobicity differences, qPCR expression patterns, histological observations) and include speculative statements about evolutionary mechanisms:

  1. Phrases like “probably evolved together” and “may be related to” are speculative; the study does not directly demonstrate evolutionary mechanisms. Conclusions should stick to what the data directly suport;
  2. Does not mention the hydrophobicity results, qPCR expression trends, or functional insights from histology, all of which were central to the results;
  3. No mention of ecological or comparative aspects (e.g., Antarctic icefish, deep-sea adaptations);
  4. The phrase “structural differences among species may be related to dissimilar adaptations and functional requirements” is vague; it could be more specific by referring to hydrophobic domains, exon length, or paralog-specific expression patterns.

The Abstract partially conveys the study’s results, but it needs stronger data support and clearer method description:

  1. The conclusion sentence (“probably evolved together with the divergence…”) is speculative and not directly demonstrated by the data;
  2. Simply stating “searching for eln genes in sequenced genomes” does not indicate methods for phylogenetic, hydrophobicity, histology, or qPCR analyses, which are major parts of the study;
  3. Mentions strong expression of elnb but does not quantify fold differences or number of samples;
  4. Hydrophobicity differences are described qualitatively; numeric ranges could improve precision.

The study’s title is appropriate but could be slightly more informative to reflect functional and physiological analyses, e.g.: “Duplication, Divergence, and Cardiac Expression of Elastin in Jawed Fishes, Including Tetraploid Rainbow Trout (Oncorhynchus mykiss).”

The references used are mostly appropriate and relevant, however:

  1. A few references (e.g., bioRxiv preprints 25, 26 and 57) are very recent or not peer-reviewed; reliance on these may weaken the foundational context;
  2. Some early references (e.g., Randall 1968; Licht & Harris 1973) are quite old and might be supplemented with more recent work on bulbus arteriosus physiology and elastin;
  3. No clear inclusion of genome-wide studies across multiple fish clades, which is central to the manuscript’s comparative genomics approach;
  4. References to extreme adaptations (snailfish, deep sea, Antarctic) are somewhat fragmented; some cited studies do not explicitly connect elastin hydrophobicity to pressure adaptation.

Author Response

Referee Report on the Manuscript: “Duplication and Divergence of Elastin in Jawed Fishes, Including Tetraploid Rainbow Trout (Oncorhynchus mykiss)” submitted to the “Genes” periodical.

n the reviewed study, the authors performed a comparative genomic, structural, phylogenetic, and expression analysis of elastin (eln) genes across 51 jawed fish species, including cartilaginous fishes, basal ray-finned fishes, teleosts, and lobe-finned fishes. The study offers several original contributions, some of which have not been previously reported, including:

  • The most comprehensive survey to date of elastin gene duplication and structure across jawed fishes
  • A new evolutionary interpretation of elna / elnb division in teleosts
  • The discovery of extreme exon amplification in the lesser devil ray elastin gene
  • Clarification of paralog retention after lineage-specific tetraploidization
  • Integration of structural protein hydrophobicity with physiological data like blood pressure
  • The first expression comparison of all three eln genes (elna1, elna2, elnb) in rainbow trout heart

Therefore, the presented findings substantially advance our understanding of elastin evolution, particularly in teleosts and tetraploid salmonids, and uncover unexpected structural diversity that may stimulate new biomechanical or developmental research.

The introduction is largely accurate in its description of elastin structure, assembly, biochemical properties, and cardiovascular function across vertebrates. Foundational statements about elastin domains, crosslinking, hydrophobicity and self-assembly are consistent with classic and modern literature. However, important evolutionary statements are oversimplified or insufficiently referenced, some claims are speculative, some terminology is misleading and the narrative is not enough tightly connected to the study’s aims:

  • the phrase “tropoelastin monomers (onwards named elastin)” is misleading, because tropoelastin and elastin are distinct: tropoelastin is the soluble precursor, and elastin is the insoluble, crosslinked polymer;

This has been corrected.

  • The statement that elastin is present “in all jawed vertebrates” needs nuance and references. It is true for most gnathostomes, but absence of bulbar elastin in Antarctic icefish (cited later) contradicts this generalization. Authors should to acknowledge known exceptions immediately, not many lines later;

Elastin is present also in Antarctic icefish, except in the bulbus.

  • The claim that the bulbus arteriosus emerged after WGD ~300 Mya is in my opinion oversimplified as current comparative anatomical literature suggests that a bulbus arteriosus-like structure exists in basal actinopterygians (gars, bowfin), not only in teleost and its evolution is not clearly tied to the teleost WGD event. This statement requires updated references or revision;

We agree and have revised this at p. 3, l. 28-33.

  • At present, the claim that duplicated eln genes “resulted in neofunctionalization of ElnB” is rather speculative as only a few studies (largely zebrafish-focused) discuss functional divergence. “Neofunctionalization” is a strong claim and should be toned down without functional studies across taxa;

We agree and have omitted “neofunctionalization” at p. 3, l. 29-31.

  • Evolutionary claims about hydrophobicity and endothermy require stronger referencing. Also, statements linking hydrophobicity to high blood pressure or coacervation temperature depend heavily on a few speculative papers. I suggest to expanded referencing and keep more cautious wording (e.g., “proposed” instead of “suggested to be an adaptive advantage”);

We agree that the first papers cited are old, but similar findings are discussed in details by Chalmers et al. 1999, which we do not consider speculative.    

  • The introduction does not clearly justify why studying elastin diversity in 51 fish is important beyond broad evolutionary interest. Specifically, missing elements include: (1) explanation of knowledge gaps (e.g., limited comparative studies of eln in non-model fishes), (2) how exon-level diversity may influence biomechanics and (3) why tetraploid species are especially informative;

This has been revised on p. 4, l. 1-10.

  • There is lack of any mention of functional studies on salmonid elastins (if available) and Clear integration of whole genome duplication theory relevant to salmonids and cyprinid;

Revised on p. 4, l. 5-10.

  • Overall, English is understandable and only requires minor editing.

The material and methods chapter is generally logical (comparative genomics + phylogenetics + histology + qPCR) and appropriate for the study’s aims. However, the description of the methods sometimes seems to be incomplete and vague:

  • No description of inclusion/exclusion criteria for selecting genomes (51 species is mentioned in the Results but not in Methods);

This has been added on p. 5, l. 3-5.

  • No explanation of genome assembly quality (many fish assemblies are fragmented or contain misannotations);

Fragmented cyprinid genomes are mentioned on p. 5, l. 13-14. We have no explanation for the low quality of the assemblies.

  • No quality control of gene predictions (manual curation? blastx? reciprocal BLAST?);

All genes and predicted proteins were controlled as described on p. 5, l. 4-14.

  • Hydrophobicity analysis lacks replicability (no sliding-window parameters, no smoothing, no amino acid scales besides Kyte–Doolittle);

The analyses are now described on p. 5, l. 25-28.

  • BLAST method not specified: BLASTP? BLASTN? TBLASTN?

Now revised on. 5, l. 6-13.

  • Search parameters missing (e-value threshold, identity %, coverage);

Default values were used

  • No description of how partial or fragmented hits were handled;

This is now briefly described on p. 5, l. 10-14.

  • No pipeline for distinguishing elnA vs. elnB paralogs;

This is now described on p. 5, l. 14-17.

  • Synteny analysis only states “adjacent limk1 and septin4” but gives no method for validating synteny (genome browser? NCBI? Ensembl?;

The synteny was described in the referred paper of Moriyama et al. (2016) and was verified in all fish eln genes p. 5, l. 9-11 and p. 7. L. 30-32.

  • Manual search for motifs” is vague and not acceptable as a reproducible bioinformatic procedure;

Repeat hydrophobic motives and crosslinks were identified by the ScanProsite tool https://prosite.expasy.org/scanprosite/ (p. 5, 29-31)

  • Presently, it is considered that NJ method is outdated. Authors should justify its use or preferably use maximum likelihood (ML) in my opinion;

We agree about using ML as described on p. 5, l. 18-24 and shown in Figure 1 on p. 8.

  • For phylogenetic analyses there is no information about alignment trimming, gaps treatment, nucleotide substitution model selection and whether elnB2 sequences were excluded a priori;

The phylogenetic analysis is briefly described on p. 5, l. 18-24.

  • For hydrophobicity analysis the link to an online calculator is insufficient for reproducibility. Missing information about window size Model justification, whether the full-length proteins were aligned or compared positionally;

Hydrophobicity analysis is described on p. 5, l. 25-30 using default values.  

  • Fish Holding and Sampling subchapter:
  • only 20 fish per timepoint are mentioned, but there is no replication structure (biological vs. technical replicates), sex information, or justification for sample size;

Gene expression was compared in bulbus and ventricle in individuals, not between sampling points. Duplicated technical qPCR replicates gave low SE values indicates sufficient number of fish per sample (p. 7, l. 23-25. Both males and females were included (p. 6, l. 6 ).

  • No explanation why specific timepoints were chosen in relation to elastin biology;

The four timepoints covered the fry, juvenile and adult stages, but the need for further studies during embryogenesis has been added in Discussion (p. 1, l. 11-2).

  • The research claims to investigate tetraploid salmonids and cyprinids, but only rainbow trout is sampled for physiology and expression;

This has been revised on p. 4 l. 26-27: “The phylogeny of eln genes was examined in multiple salmonids and cyprinids”.

  • temperature, oxygen, light regime too coarse; e.g., “constant light” is inappropriate for salmonids and should be justified;

This is now described on p. 6, l. 1-13.

  • Growth conditions (tank densities, feeding rate, water flow) are missing;

This is now described on p. 6, l. 1-13.

  • No description of heart dissection protocol, RNA-later incubation time, or tissue storage temperaturÄ™;

Now described on p. 6, l. 21-24.

  • The larvae were “1 g” – rather “1 g fry”, not larvae;

We acknowledge this correction.

  • Transport, stress handling, and acclimation periods before sampling not described.

The fish were sampled, anaesthetized and killed before removing heart at the facilities of the rainbow trout breeding company “Svanøy Havbruk” as described in Mat. Meth.

  1. Histology subchapter missing details on:
  • No mention of the number of sections analyzed per fish, the anatomical orientation, or blinding of histological assessments
  • Fixation time in paraformaldehyde;
  • Orientation of embedding;
  • Sectioning plane;
  • Image acquisition settings (magnification, camera, scale bars);
  • Staining controls

The description of histology analysis has been revised on p. 6. l. 30-32 and p. 7, l. 1-4.

  1. Molecular Analysis subchapter missing details on:
  • Subchapter title – better fits gene expression analysis or just “Real-time PCR analysis”;
  • RNA integrity not assessed;
  • No description of DNAse treatment;
  • SYBR Green qPCR conditions are oversimplified and lack of exact replicate numbers (technical & biological);
  • There is no explaination of statistic test selection and checking of data distribution and variance homogeneity.

The description of gene expression analysis has been revised on p. 7, l. 5-25.

The results appear generally clear, but some sections could benefit from more explicit highlighting of trends, clearer linkage between figures/tables and text, and more precise explanations of key comparisons:

  • Paragraphs frequently mix results with interpretation;

This has been improved in the revised version.

  • Missing clarity about assembly quality (which influences gene detection);

All genome assemblies seem to be of high quality, although some partial eln sequences are given in Supplementary Table S1.

  • Statements such as “considerably larger” are qualitative and actual exon lengths should be reported;

Exon lengths have been included in the given examples, such as yellowfin tuna eln genes (p. 9, l.10-12).

  • No statistical values shown for gene expression analysis results (fold change, p values);
  • The statement “The bulbus and ventricle were not dissected in the larvae” appears misplaced and breaks narrative flow;

This has now been deleted.

  • “up to 15 times higher” is vague—use exact fold-change per timepoint;

This has been revised (p. 10, l. 25).

  • Claims like “varies largely” or “highly hydrophobic elastin was consistently found” need statistical support;

This has been revised and statistics have been included on p. 11, l. 16-19.

  • Hydrophobicity values should be given as mean ± SD;

This has been corrected (p. 11, l. 17-18).

  • It is unclear whether analyses used full-length sequences, ortholog alignments, or paralog-specific datasets;

Only full-length sequences were analysed (p. 5, l. 10).

  • Relationship with blood pressure – no numeric correlation coefficient (r, p-value);

This has now been added

  • “In comparison, the tetrapod species examined possessed high elastin hydrophobicity and high blood pressure levels.” → observational but should be supported visually and numerically;

This is shown in Figure 7.

  • No statistical test described for relationship with blood pressure (linear regression? Spearman?).

Relationship was examined by linear regression.

The Discussion requires restructuring, clarification, and moderation of speculative claims:

  • Statements like “high distensibility and resilience of the bulbus have consistently been reported in various teleost species” are broad. No specific numbers or citations for all species mentioned;

The studies of three teleosts have been cited (p. 18, l. 14), and physiological implications for the blood flow in rainbow trout is cited (p. 18, l. 16-17) 

  • Claims about snailfish Glu/Asp content affecting vessel stiffness is rather speculative as no functional or experimental data supports this;

We agree and have deleted this paragraph.

  • Linking hydrophobicity directly to blood pressure across species is overstated, given results showed no correlation;

We do not understand the comment as our study aimed to finally disprove the proposed relationship between elastin hydrophobicity and blood pressure.

  • The functional significance of Antarctic icefish adaptations is partially conjectural without experimental validation;

We do not understand this comment, since we have cited well-documented papers by reputed scientists studying Antarctic fish for decades.

  • Paragraphs mix multiple topics (e.g., gene divergence, expression patterns, hydrophobicity, ecological adaptations) without smooth transition;
  • The flow jumps between rainbow trout, Antarctic fish, teleosts, and mammals abruptly, making the argument hard to follow;

These paragraphs have been improved in the revised version.

  • Use of “athletic tuna fish has thick aortic elastin lamella and bulbar elastin fibres” lacks quantification.

We have searched the literature without finding quantitative measurements of the elastin fibers in tuna and zebrafish.

  • The conclusions are partially supported by the results: they correctly note the conservation of non-teleost elastin and divergence of teleost ElnA/ElnB, but they omit key findings (e.g., hydrophobicity differences, qPCR expression patterns, histological observations) and include speculative statements about evolutionary mechanisms:

The conclusions have been completely revised according to the valuable comments below.

  • Phrases like “probably evolved together” and “may be related to” are speculative; the study does not directly demonstrate evolutionary mechanisms. Conclusions should stick to what the data directly suport;
  • Does not mention the hydrophobicity results, qPCR expression trends, or functional insights from histology, all of which were central to the results;
  • No mention of ecological or comparative aspects (e.g., Antarctic icefish, deep-sea adaptations);
  • The phrase “structural differences among species may be related to dissimilar adaptations and functional requirements” is vague; it could be more specific by referring to hydrophobic domains, exon length, or paralog-specific expression patterns.

The Abstract partially conveys the study’s results, but it needs stronger data support and clearer method description:

The Abstract has been revised as suggested below.

  1. The conclusion sentence (“probably evolved together with the divergence…”) is speculative and not directly demonstrated by the data;
  2. Simply stating “searching for eln genes in sequenced genomes” does not indicate methods for phylogenetic, hydrophobicity, histology, or qPCR analyses, which are major parts of the study;
  3. Mentions strong expression of elnb but does not quantify fold differences or number of samples;
  4. Hydrophobicity differences are described qualitatively; numeric ranges could improve precision.

The study’s title is appropriate but could be slightly more informative to reflect functional and physiological analyses, e.g.: “Duplication, Divergence, and Cardiac Expression of Elastin in Jawed Fishes, Including Tetraploid Rainbow Trout (Oncorhynchus mykiss).”

We really acknowledge the suggested title, which has now been revised.

The references used are mostly appropriate and relevant, however:

  • A few references (e.g., bioRxiv preprints 25, 26 and 57) are very recent or not peer-reviewed; reliance on these may weaken the foundational context;

We have decided to leave the decision about these references to the Editor.

  • Some early references (e.g., Randall 1968; Licht & Harris 1973) are quite old and might be supplemented with more recent work on bulbus arteriosus physiology and elastin;

The five references are spanning 35 years and are central to this topic. The crucial role played by teleost ElnB is described in the same paragraphs, including relevant references.

  • No clear inclusion of genome-wide studies across multiple fish clades, which is central to the manuscript’s comparative genomics approach;

Genome-wide study of duplicated genes in salmonids has been included (p. 4, l. 6-7).

  • References to extreme adaptations (snailfish, deep sea, Antarctic) are somewhat fragmented; some cited studies do not explicitly connect elastin hydrophobicity to pressure adaptation.

We agree about this, but the cited studies exemplifies the diversity in fish habitats and adaptations that motivated our study and should stimulate further research.

Reviewer 2 Report

Comments and Suggestions for Authors

In general, it is an interesting and well done study. However, there are some points which should be corrected. There are some too simplicised phrase about the evolution, especially concerning the terms primitive used for organism, not a structure, and rather popular or old-fashioned understanding of fishes - now we know that there are several classes, there is no a monophyletic taxon whose English name would be "fish". Please see my comment on the file. However, there is one rather important problem: the presented phylogenetic tree cannot be accepted. First of all, it makes no sense to infer such "phylogeny", presenting the "phylogenetic relationships" between two molecules found in the same specimen. These should be either two trees, or one, but computed for concatenated two sequences in each mOTU. Secondly, neighbor-joining is very useful for computing an initial tree, or for some preliminary analysis, but certainly not for publication: some maximum likelihood program should be used. Latin names should be used in the tree, etc. Again, please read my comments in the file. An the tree is graphically not good enough.

Author Response

In general, it is an interesting and well done study. However, there are some points which should be corrected. There are some too simplicised phrase about the evolution, especially concerning the terms primitive used for organism, not a structure, and rather popular or old-fashioned understanding of fishes - now we know that there are several classes, there is no a monophyletic taxon whose English name would be "fish". Please see my comment on the file. However, there is one rather important problem: the presented phylogenetic tree cannot be accepted. First of all, it makes no sense to infer such "phylogeny", presenting the "phylogenetic relationships" between two molecules found in the same specimen. These should be either two trees, or one, but computed for concatenated two sequences in each mOTU. Secondly, neighbor-joining is very useful for computing an initial tree, or for some preliminary analysis, but certainly not for publication: some maximum likelihood program should be used. Latin names should be used in the tree, etc. Again, please read my comments in the file. An the tree is graphically not good enough.

We agree about the term “phylogeny”, which has been corrected to evolutionary tree.

The NL method has been replaced with ML.

Reviewer 3 Report

Comments and Suggestions for Authors

In this manuscript, Anderson and Ostbye probe the NCBI database to compare the diversity of elastin paralogs across major fish clades. In teleosts the additional genome duplication has conserved both ElnA and ElnB (up to 4 total) genes, with ElnB paralogs encoding longer hydrophobic domains, which has been proposed to decrease stiffness of the bulbus arteriosus. However, the authors find no correlation between hydrophobicity and ventral aortic blood pressure.

The manuscript is clearly written but there are a few issues that could be addressed to improve the study.

1) In the one experimental example provided, which is the rainbow trout, the authors claim that ElnB expression dominates the BA. This is somewhat expected based on known expression patterns of paralogs in zebrafish, but they do not really show this. The larval stain showing elastin in the larval BA and aorta (Fig. 4) does not discriminate isoforms. The qPCR data (Fig. 5) appears to show about similar total amounts of A1/2 as B1 transcripts. It would be a very nice and a much more convincing strategy to use in situ hybridization for the 3 transcripts in larvae. This is a very standard protocol but if the investigators might not be equipped, perhaps a collaborator could assist?

2) The authors state that ElnB paralogs differ primarily due to having longer hydrophobic domains compared to ElnA. Yet the ElnB proteins show relatively less hydrophobicity (Fig. 6)? Can this be clarified? Does this explain why there is no correlation with blood pressure?

3) Some of the data points in Fig. 7 are missing or mis-labelled. For example, I do not see on the plot Pe.ma, Sc.ca, Mo.hy, Pr.pe, etc.

Author Response

In this manuscript, Anderson and Ostbye probe the NCBI database to compare the diversity of elastin paralogs across major fish clades. In teleosts the additional genome duplication has conserved both ElnA and ElnB (up to 4 total) genes, with ElnB paralogs encoding longer hydrophobic domains, which has been proposed to decrease stiffness of the bulbus arteriosus. However, the authors find no correlation between hydrophobicity and ventral aortic blood pressure.

The manuscript is clearly written but there are a few issues that could be addressed to improve the study.

1) In the one experimental example provided, which is the rainbow trout, the authors claim that ElnB expression dominates the BA. This is somewhat expected based on known expression patterns of paralogs in zebrafish, but they do not really show this. The larval stain showing elastin in the larval BA and aorta (Fig. 4) does not discriminate isoforms. The qPCR data (Fig. 5) appears to show about similar total amounts of A1/2 as B1 transcripts. It would be a very nice and a much more convincing strategy to use in situ hybridization for the 3 transcripts in larvae. This is a very standard protocol but if the investigators might not be equipped, perhaps a collaborator could assist?

We acknowledge the comment about and have revised this part at p. 10, l. 33-37).

Regarding the strategies to study the spatial and temporal localization of the three eln genes, this will be examined in another study as also suggested in Discussion (p. 14, l. 10-12).

2) The authors state that ElnB paralogs differ primarily due to having longer hydrophobic domains compared to ElnA. Yet the ElnB proteins show relatively less hydrophobicity (Fig. 6)? Can this be clarified? Does this explain why there is no correlation with blood pressure?

This should be solved by the revised description of rainbow trout ElnB on p. 10, l. 3-7 and the Supplementary Figure S2.

Good question, but ElnB is only found in teleosts.

3) Some of the data points in Fig. 7 are missing or mis-labelled. For example, I do not see on the plot Pe.ma, Sc.ca, Mo.hy, Pr.pe, etc.

Thank you so much. This has now been corrected.

Round 2

Reviewer 1 Report

Comments and Suggestions for Authors

Referee Report on the Manuscript: “Evolution, Structure, and Cardiac Expression of Elastin Genes in Jawed Fishes” submitted to “Genes” periodical.

The revised manuscript demonstrates substantial improvement compared to the previous version. The authors have successfully addressed earlier concerns. The revised manuscript presents a clear, well-structured study with robust methodology, valid results and thoughtful discussion, it requires only minor editorial corrections before publication.

Author Response

The reviewer states that we have successfully adressed earlier concerns and that only minor editoral corrections are required before publication.

We have noted Ref3 comment that the language is fine and does not require any improvement.

Reviewer 3 Report

Comments and Suggestions for Authors

It is improved and recommended for publication. 

Author Response

We notice that this reviewer has recommended the revised manuscript for publication. 

Also the English is fine and does not require any improvement.